# Transformation of Corn Stover into Furan Aldehydes by One-Pot Reaction with Acidic Lithium Bromide Solution

**DOI:** 10.3390/ijms232314901

**Published:** 2022-11-28

**Authors:** Meixiang Gao, Qi Xin, Wan Sun, Jiaqi Xiao, Xianqin Lu

**Affiliations:** 1Key Laboratory of Pulp and Paper Science & Technology of Ministry of Education/Shandong Province, Qilu University of Technology (Shandong Academy of Sciences), Jinan 250353, China; 2Advanced Research Institute for Multidisciplinary Science, Qilu University of Technology (Shandong Academy of Sciences), Jinan 250353, China; 3School of Bioengineering, Qilu University of Technology (Shandong Academy of Sciences), Jinan 250353, China; 4State Key Laboratory of Microbial Technology, Shandong University, Qingdao 266237, China

**Keywords:** furfural, 5-HMF, corn stover, acidic LiBr treatment, THF

## Abstract

Currently, the production of furan aldehydes from raw biomass suffers from low furfural yield and high energy consumption. In this study, a recyclable and practical method was explored for the preparation of furfural from corn stover by the one-pot reaction by acidic lithium bromide solution (ALBS) without pretreatment and enzymolysis. In the ALBS reaction, the furan aldehydes were generated by the degradation of lignocellulose; however, the products were unstable and were further dehydrated to form humins. So, dehydration reaction was inhibited in this study, and the high yield of furan aldehydes was obtained, in which 2.94 g/L of furfural and 2.78 g/L of 5-hydroxymethyl furfural (5-HMF) were generated with high solid loading (10 wt%), the presence of commercial catalyst ZSM-5 and co-solvent tetrahydrofuran (THF) at 140 °C for 200 min. Via this method, almost 100% of hemicellulose was transformed to furfural, and 40.71% of cellulose was transformed to 5-HMF, which was based on the theoretical yield of HMF (8.35 g) from glucose (29.30 g) produced from cellulose. After the reaction, the catalyst ZSM-5 was the main component in the solid residue and kept a suitable performance. THF azeotrope was easily separated from the slurry by evaporation. During the removal of THF, lignin was precipitated from the liquid phase and showed lower molecular weight and abundant active groups, which was a potential feedstock for producing valuable aromatics and polymers. Thus, in a one-pot reaction, the ideal yield of furan aldehydes from raw biomass was obtained on a lab scale, and the catalyst, THF, and LiBr were easily recycled, which provided an option to realize the economical production of sustainable furan aldehydes from raw biomass.

## 1. Introduction

Furfural, as the promising platform chemical, could be used to synthesize various derivatives, such as furfuryl alcohol, methyl tetrahydro furan, and furan, and fuel products, such as gasoline, jet fuel, fuel blending components, bioplastics, etc. [1,2,3]. Lignocellulose was an ideal bioresource used to produce furan aldehydes, with low-price, readily available, and abundant reserves [4]. Cellulose, hemicellulose, and lignin are the main components of lignocellulose. Xylan is the main component of hemicellulose, which is easily degraded during moderate-severity acidic treatment [5,6]. Currently, the industrial furfural production process is based on a H_2_SO_4_-catalyzed reaction at temperatures ranging from 153 °C to 240 °C. During the reaction, the hemicellulose was first degraded to pentose. Then, pentose was further dehydrated into furfural [6,7]. The theoretical yield of pentose to furfural is 0.73 kg/kg pentose [8]; however, the yield of furfural in industrial practice is only about 50–55% of the theoretical yield [4,9,10,11]. The demand for furfural and furfural derivatives would increase since they act as an industrial chemical platform to produce renewable chemicals and biofuels. Thus, more efforts are needed to develop efficient processes to improve the yield of furfural from biomass sugar streams to make this pathway feasible.

Great efforts were made to improve the furfural yield from sugars and decent furfural yield was already obtained in many researches. For instance, Gao et al. reported 82.4% furfural yielded from hemicellulose in water/toluene biphasic system with ZSM-5 as catalyst [12]. Recently, 53% of furfural was also reported to be yielded from D-xylose with an aqueous HCl-1,2-dichloroethane biphasic reaction mixture using benzyltributy-lammoniu chloride as a phase transfer catalyst [13]. From that, the transformation of model sugar to furfural was practiced. However, there are still many shortcomings in conversing natural biomass to furan aldehydes, because of the impurities and natural resistance in biomass [14]. Although the conversion of raw biomass to furan aldehydes could be conducted in lab, the current process suffers from many problems, including the low biomass loading (2 wt% to 5 wt%), specific complex catalyst, toxic and expensive organic solvents, severe heating treatment, and high requirement for substrate (such as the smaller particle size of corn stover) [4,11,15,16]. For example, by using pulp prehydrolysate derived from prehydrolysis of wood, 10 h H_2_SO_4_-treatment was needed to obtained relatively high yield of furfural (78.6%) [17]. The highly energy intensive and relatively low furfural yield (vs. model sugar) prevented the commercialization of transformation of raw biomass to furan aldehydes. Therefore, it needed to develop effective strategy to fractionate raw biomass and obtain high-yields products directly from raw biomass without complex process and expensive toxic reagent. Herein, we present an integrated green process by using acidic LiBr and biomass-derived solvents with moderate treatment severity.

The acidic LiBr treatment, as a low toxicity and recyclable strategy, was often used to hydrolyze polysaccharide to monosaccharide without pretreatment and enzymolysis [18]. Pan et al. reported that furfural and 5-HMF could be detected in LiBr reaction by extending residual time or improving temperature [19]. However, the recent studies have shown that much humins (as biochar) was synthesized during acidic LiBr reaction along with the loss of fermentable sugars [19]. As reported before, during acidic reaction of lignocellulose, the furan aldehydes were the precursors during the humins formation process, which were easily crosslinked with glucose and xylose, or self-polymerized to form undesired products, such as humins [1,20,21]. Thus, LiBr-biphasic reaction was performed by replacing part of the aqueous phase with organic solvents in this study, to suppress the formation of humins, as showed in Figure 1. The furfural was produced in aqueous phase and the product was extracted to the organic phase, because of the high solubility of furfural in organic phase [22,23]. The extraction capability of organic phase was able to protect the furan aldehydes from dehydration reaction and forming humins [1,24]. In this study, the LiBr-biphasic reaction condition was optimized and high yield of the furfural and 5-HMF were produced in high solid loading with industrialized catalyst ZSM-5 zeolite.

## 2. Results and Discussion

### 2.1. Production of Furfural and 5-HMF during Biochar Production from Corn Stover

The yield of furan aldehydes (furfural and 5-HMF) was detected as showed in Table 1, which showed that the furan aldehydes (furfural and 5-HMF) were formed during acidic LiBr treatment of raw lignocellulose, however, the yields were poor. During acidic LiBr treatment in 0.1 M HCl with 140 °C for 120 min, only 0.91% of furfural yield was detected, as well as 50.89% xylose was yielded as reported before [22]. Raising the treatment intensity by increasing acid dosage to 0.5 M HCl and prolonging the residual time to 150 min was able to slightly enhance the yield of furfural to 1.72% with the loss of the sugars. Almost no saccharides were detected in the supernatant. Meanwhile, a large amount of humin was produced, however, the humin was formed and mixed with the residual solid to make the residues in black color, so it is hard to quantify the amount of humins [22]. During hydrolysis of lignocellulose, hemicellulose was easily depolymerized to xylose and the products were stably existed in mild reaction. The xylose could be further dehydrated into furfural with increasing the treatment intensity. However, humin was formed by self-polymerization of furan aldehydes or the cross-polymerization of furan aldehydes and saccharides during reaction [1,22]. Because of furan aldehydes as the precursor in humins forming, inhibiting the formation of humins was supposed to lead to the high furan aldehydes yield by adding organic solvent into the LiBr aqueous as a biphasic reaction.

### 2.2. Conversion of Corn Stover to Furfural by Inhibiting the Humins Formation

A biphasic reaction was carried out by adding isovolumetric organic phase N,N-Dimethylformanmide (DMF) into the acidic LiBr treatment process, as shown in Table 2 [25]. It showed that the presence of DMF distinctly increased the yield of furan aldehydes to 31.17% for furfural and 17.15% for 5-HMF with 0.5 M HCl at 140 °C for 150 min. During the reaction, the addition of commercial catalyst ZSM-5 can further enhance the yield of furfural to 36.00%. As well, the production of 5-HMF was increased to 19.40% with the presence of ZSM-5. The Si/Al molar ratio for ZSM-5 used in this study is 40. The results suggested that the ZSM-5 could be in selectivity to furfural and 5-HMF, where xylose was conversed to furfural by taking off three molecules of water under the action of Brønsted acid sites offered by ZSM-5 and H+ in this system. As well, the intermediate enol structure was produced from glucose on the Lewis acid active sites offered by ZSM-5 in this system and further hydrolyzed to 5-HMF on the Brønsted acid sites offered by ZSM-5 and H+ in this system. However, only a 4.36% yield of furfural and a 3.05% yield of 5-HMF were obtained in the mild reaction condition (140 °C for 120 min and 0.1 M concentration of HCl) with DMF and ZSM-5. As described above, a high yield of xylose was obtained, and it could be stable in a mild reaction [22]. It indicated that the severity of the reaction, including high acid dosage and strong heat treatment, played a distinct role in the dehydration of sugars to form furan aldehydes.

### 2.3. Changing the Reaction Conditions for High Yield of Furan Aldehydes

In order to obtain the high yield of furfural and 5-HMF, the addition of LiBr (1.5 mL, 2.5 mL, 3 mL, 4 mL, and 5 mL), the substrate loading (5 wt%, 10 wt%, and 20 wt%), the types of the organic phase (DMF, GVL, DMSO, and THF), the addition of organic phase (5 mL, 10 mL, 15 mL, and 20 mL), and the reaction time (120 min, 150 min, 170 min, 200 min, and 220 min) were discussed in the following.

The concentrated LiBr was able to disrupt the hydrogen bond in cellulose, swell the crystalline cellulose, and dissolve the cellulose in the reaction system, which was the essential factor in the saccharification process of raw lignocellulose. So, the LiBr concentration was discussed first. By decreasing the addition of LiBr solution, the production of furfural was reduced to 1.47–0.98% as test 1-1 to 1-4 in Table 3. Meanwhile, almost no 5-HMF was detected in that reaction. It suggested that the concentration of LiBr was the important factor affecting the liquidation of raw lignocellulose. In the reaction, high LiBr concentration led to the formation of un-solvated Li ions and naked Br ions. The ions were able to disrupt the hydrogen bonds in cellulose and promote the dissolution of feedstock [26]. Pan et al. reported that 60 wt% LiBr was appropriate to liquefy the raw lignocellulose in which cellulose was completely hydrolyzed and 85% soluble saccharides were generated [26]. However, further increasing the concentration of LiBr in the reaction led to the decrease in dissolubility to cellulose because of the insufficiency of water in the reaction [26]. A total of 0.6 g LiBr/mL solution showed well performance in the dissolution of corn stover in our previous study [19,22]. Because of the dissolving capacity of LiBr in organic and aqueous phases, a 1.2 g/mL LiBr aqueous phase was elected in this reaction. In addition, the 1.2 g LiBr/mL solution was almost the maximum solubility for LiBr aqueous solution at the melt temperature (about 50 °C), and it was selected for further study.

The loading of the substrate, as the solid loading, was adjusted in the specific LiBr concentration to make a relatively high yield of furfural and 5-HMF. The substrate loading of 5 wt% and 20 wt% were compared in the text as test 2-1 to 2-2, shown in Table 3. It was obviously observed that increasing the solid loading decreased the yield of furan aldehydes, in which only 19.65% furfural and 9.01% 5-HMF were yielded at the 20% solid loading. Meanwhile, at 5 wt% substrate loading, the yield of furan aldehydes was increased, especially for 5-HMF with a 22.00% yield. Lowering solid loading was a way to increase the conversion ratio of biomass to furan aldehydes, which was why much research was performed with low biomass loading (2 wt% to 5 wt%) [4,9,11,15,16]. However, it showed that only 0.08 g/L furfural and 0.13 g/L 5-HMF were produced in the reaction with 5 wt% substrate loading. A total of 0.18 g/L furfural and 0.21 g/L 5-HMF were yielded during LiBr treatment with 20 wt% substrate loading. Due to the low-price, readily available, and abundant reserves of raw biomass, high concentration of furan aldehydes yields in the reaction slurry was more catering possibility to commercial applications [4]. In this study, 0.16 g/L furfural and 0.22 g/L 5-HMF were obtained with 10 wt% solid loading. By comparison, 10 wt% substrate loading was more practical.

The optimal reaction condition was carried out, and cellulose and hemicellulose were able to be saccharified to produce glucose and xylose, then the monosaccharides were further dehydrated to form furan aldehydes. It is known that, during the reaction, the water solvent shell was crucial in the dehydration reaction of glucose to form the furan aldehydes [27,28]. However, excessive water was detrimental to furan aldehydes selectivity, and the undesired products such as humins were enhanced [9,24]. The co-solvent could increase the furan aldehydes selectivity by extracting the production from the aqueous phase [29]. So, the types and the addition of organic phase were assessed. The different types of organic solvent γ-valerolactone (GVL), tetrahydrofuran (THF) and dimethyl sufoxide (DMSO) were introduced in reaction as test 3-1 to 3-3 showed in Table 3. It observed that the furan aldehydes selectivity in LiBr reaction was decreased in the order of THF, DMF > GVL > DMSO. A total of 38.10% furfural yield and 19.64% 5-HMF yield were obtained with the present of THF, which performed as well as that in DMF. Although the GVL, DMSO, DMF and THF were used as co-solvent in previous researches, the poor selectivity of expensive GVL and toxic DMSO on furan aldehydes in this study hindered their performance [12,24]. With the present of DMSO, the amount of humins in black as biochar were formed after reaction. During the reaction, co-solvent DMF performed well, but the high boiling point (158 °C) of DMF made it hard to move off after reaction. THF showed many advantages as the co-solvent such as non-toxicity, low boiling, and recoverability, thus it was used in the following.

The ratios of the organic solvent-aqueous phase were set as 1:1, 2:1, 3:1, and 4:1 (THF to LiBr aqueous phase, *v/v*) in the biphasic reaction. It showed that (test 4-1 to 4-3 in Table 3) increasing the solvent-aqueous phase rate from 1:1 to 3:1, the furfural was yielded from 38.10% to 74.49% and the yield of 5-HMF was from 19.64% to 34.94%, while the productions yield (66.68% furfural yield and 27.17% 5-HMF yield) were reduced with 1:4 solution. In this study, a 3:1 (*v/v*) solvent-aqueous phase reaction was optimal for the formation of furan aldehydes.

At last, the appropriate reaction time was discussed to make a higher yield of furfural and 5-HMF, and the lower production of humins is shown in Table 3. It showed that by prolonging the reaction time to 200 min, the furfural yielded was further increased to 50.26%, showing as 5-1 to 5-2. However, further increasing the reaction temperature to 220 min, the furfural yield was decreased, and the humins were observed during the reaction. From that, the optimum condition was determined, as 10 wt% raw corn stover loading with catalyst ZSM-5 and co-solvent THF in 3:1 (*v/v*), at 140 °C for 200 min. Under that approach, the highest furan aldehydes concentration was 2.94 g/L furfural and 2.78 g/L 5-HMF, in which almost 100% of hemicellulose was transformed to furfural, and 40.71% of cellulose was transformed to 5-HMF.

In a previous study, corn stover was used as the feedstock to produce furan aldehydes, as shown in Table 4. Zhu et al. reported that a 68.6% yield of furfural was obtained from corn stover with a specific carbon-based heterogeneous catalyst and a high residual temperature of 200 °C [30]. Zhang et al. reported that 60.6% of the yield of furfural was produced from corn stover with carbonaceous material as a catalyst [31]. Nowadays, many catalysts are synthesized and used in the conversion process of corn stover to produce 5-HMF and furfural, and a high yield of products was obtained, as shown in Table 4. Such as, Li et al. explored the conversion of biomass (corncob, corn straw, and eucalyptus sawdust) and xylose to furfural with sulfonated carbon microspheres catalyst, and the results showed more than 70% and 75.12% furfural were produced from biomass and xylose, respectively [32]. With glucose as the substrate, 52.9% 5-HMF yield was reported by Zuo et al., with the 2% solid loading for substrate in the ChCl-H_2_O system at 120 °C for 120 min [33]. Teng et al. reported 77.82% of furfural and 33.20% of 5-HMF were yielded from corn stover; however, the reaction was performed with a low solid loading of biomass to 2 wt%, high addition of 10 wt% specific catalysts produced from the carbonized rape pollen with SO_4_^2−^/Sn, and strong heating treatment at 190 °C for 3 h [16]. The maximum furfural yield reached 95.1%, reported by Li et al., but the reaction was performed with 0.3 g corn stover and 0.3 g specific catalysts (SAPO-18 zeolites) in 12 mL GVL and 3 mL aqueous phase at 205 °C for 40 min [4]. Yang et al. studied that 53% furfural and 17.0% of 5-HMF were yielded with the catalyst SO_3_H-NG-C in the presence of GVL with only 1.3% solid loading and a high reaction temperature of 190 °C [34]. Lai et al. also reported a relatively high yield of furfural with 53.0% and 5-HMF at 25.6% in the H_2_O-MIBK system with 3.3% solid loading at 170 °C for 180 min [35]. By comparison, the method carried out in this study not only obtained a high yield of 5-HMF and furfural with corn stover as the feedstock but also did perform in high solid loading (10%) for substrate and relatively low reaction temperature. Furthermore, no matter the cost or the preparation process, the ZSM-5 catalyst adopted in this work was obviously superior to C-Co-S, AlCl_3,_ SO_3_H-NG-C, Ch15-AgPW, SO_4_^2−^/Sn-TRP, and SAPO-18 catalysts. The running economy of the catalyst is more conducive to deep-level research into mechanism and reaction dynamics, which will promote progress in industrialization. From that, it suggested that it was an effective strategy to produce furan aldehydes from corn stover by one-pot reaction with acidic lithium bromide solution, which obtained high-yield products directly from raw biomass without complex process, expensive toxic reagent, and in moderate treatment severity.

### 2.4. The Properties and Reusability of Catalyst and Lignin after Reaction

Industrialized catalyst ZSM-5 zeolite is chosen as the catalyst in this method because of its several advantages, including its low price and high catalysis efficiency. There are many Brønsted sites in ZSM-5, which can promote the conversion of xylose to form furfural and glucose to 5-HMF. The properties of catalysts play an important role in the catalysis process. During the optimal reaction condition, the corn stover was almost completely hydrolyzed, and the catalyst ZSM-5 was the main component in the solid residue. We collected the residual catalyst ZSM-5, and the residue was directly characterized by TEM (transmission electron microscopy), FTIR, TPD of NH3, and BET (Brunauer–Emmett–Teller). It showed that the morphology of ZSM-5 was in a round ball, and it retained its morphology after the reaction, as shown in Figure 2a. The surface area, pore size, and total acid content of ZSM-5 were slightly changed, as shown in Table 5. The FTIR spectrum showed that the C=O stretching vibration at 1383 cm^−1^ was reduced in ZSM-5 after the reaction, and no obvious changes were observed in Si-O stretching vibration at 448 cm^−1^ and 802 cm^−1^, in the signals of Si-O-Al at 550 cm^−1^ and the signals at 1104 cm^−1^ for Si-O-Si, in Figure 2b. The adsorption of some biochar on the ZSM-5 surface may be the reason for the decrease in C=O stretching vibration, which could be recovered by desorption. In this study, the reusability of ZSM-5 has been conducted, in which ZSM-5 catalyst after the reaction was collected and roasted in a tube furnace at 550 °C for 3 h. The catalyst was reused in the reaction of corn stover into furans, and the yield of products showed in Table 3. It suggested that the properties of the ZSM-5 catalyst kept in a suitable performance, making the recyclable potential of ZSM-5. 

After the separation of ZSM-5 by the filter, THF azeotrope was easily separated from the slurry by evaporation. During the acidic LiBr treatment of lignocellulose, the polysaccharides were depolymerized by the catalysis of LiBr. Lignin is resistant to digestion, but it is able to be dissolved in the THF solution during the reaction [26]. With the removal of THF, lignin was gradually precipitated from the liquid phase due to its dissolubility to water, and it was separated as a bottom residue [10]. The FTIR spectrum of lignin, shown in Figure 3, was similar to the sulphuric acid extracted lignin, as described by Cai et al. [10]. However, it obtained a stranger absorbance band at the carbonyl stretching and OH stretching vibration in R-OH and Ar-OH. As well, the GPC analysis showed that the molecular weight of lignin was distinctly reduced, in which the Mn (the number average molar mass) to residual lignin was 1,138 Da, and that was 19,070 Da for the lignin in raw corn stover [36]. It indicated that the C-C linkages in lignin were also cleaved in the presence of H^+^ and the high concentration of LiBr [18,37]. Then, the cleaved C-C linkages were reduced to Hibbert’s ketones and benzodioxanes, which provided more reaction sites on the residual lignin surface [37,38]. Thus, the residual lignin showed a high re-dissolving capability to THF with low molecular weight and abundant chemical active sites, which was a potential feedstock used to upgrade to valuable aromatics and polymers [39,40].

Based on the above results, the mechanism of the ALBH process was discussed. Briefly, in this method, the high concentrated LiBr was able to swell and dissolve the cellulose and hemicellulose in the system. Then the acid-catalyzed hydrolysis was carried out with the presence of HCl, in which glucose and xylose were produced. Xylose was conversed to furfural by taking off three molecules of water under the action of Brønsted acid sites, which was offered by ZSM-5 and H^+^ in this system. The intermediate enol structure was produced from glucose on the Lewis acid active sites, which was offered by ZSM-5 in this system, and further hydrolyzed to 5-HMF on the Brønsted acid sites offered by ZSM-5 and H^+^ in this system [7,41,42]. The extraction capability of the organic phase was able to protect the furan aldehydes from dehydration. However, the residual furfural and 5-HMF in the aqueous phase were dehydrated to humins. Some humins were deposed on the reaction sites of the catalyst, leading to the deactivation of the catalyst. To reuse the catalyst, the ZSM-5 catalyst, after the reaction, was roasted in a tube furnace at 550 °C for 3 h and was reused for the reaction of corn stover into furan aldehydes.

### 2.5. Integrated Protocol to Apply the Method in Biorefinery

Figure 1 illustrates the integrated protocol to apply the method in biorefinery. We proposed the raw corn stover as the feedstock, and commercial catalyst ZSM-5 and acidic LiBr aqueous were added to the reactor. The reaction proceeded at 140 °C for 200 min. After the reaction, most of the corn stover was dissolved in the solution, and the ZSM-5 catalyst could be sent directly to the next reactor. The residual THF and the products of furan aldehydes stayed in the liquid phase and as water-volatile substances, which were easily separated from the slurry by evaporation [7]. The THF azeotrope was firstly separated from the slurry at 30 °C in vacuum, and only no more than 4.6 wt% of water was contained, so the recovered THF could be recycled without purification. During the removal of THF, lignin was precipitated from the liquid phase, and it was separated as a bottom residue by centrifugation. The furan aldehydes were recovered in the following by evaporation with a boiling point of 97.9 °C [10]. At last, the acidic LiBr was the main component in the residual solution, and it could be recycled directly to a second reactor. 

Thus, during the transformation of corn stover into furans by one-pot reaction with acidic lithium bromide solution, a biphasic reaction by replacing part of the aqueous phase with organic solvents was a way to reduce the formation of humins. Because of the high solubility of furfural in the organic phase, the furfural was produced in the aqueous phase, and the product was extracted to the organic phase [23]. The extraction capability of the organic phase was able to protect the furan aldehydes from the dehydration of furfural or 5-HMF and decomposition to humins [1,24]. By calculating the mass balance of the reaction, it suggested that 100 g corn stover is able to produce 7.86 g furfural and 8.35 g of 5-HMF eventually. The detail of mass balance is shown in the following. Briefly, 100 g corn stover is composed of 13.97 g hemicellulose, 32.56 g cellulose, and 17.14 g lignin. The carbohydrate (46.53 g, including cellulose and hemicellulose) is able to be hydrolyzed into 12.29 g xylose, 29.30 g glucose, and 10.72 g soluble oligosaccharide, respectively. Consequently, 7.86 g (yield of 100%) furfural and 8.35 g of 5-HMF (yield of 40.71%) were obtained by acid catalyst treatment. This reaction has no requirement on the particle size of the substrate; thus, any pretreatment for the raw biomass, including presoak and hydrolysis, was not needed. Moreover, in a one-pot reaction, decent furan aldehydes yield from raw biomass was obtained on a lab scale. It may realize the economical production of sustainable furan aldehydes from raw biomass by integrating this method with recent developments in furan aldehydes separation methods.

## 3. Materials and Methods

### 3.1. Materials

Corn stover was collected from the farmland in Liaocheng (Shandong Province, China). The pulverizer with a screen was used to crush the raw corn stover. The composition of corn stover was analyzed by the National Renewable Energy Laboratory (NREL) methods, with 32.56 wt% of cellulose, 13.97 wt% of hemicellulose, and 17.14 wt% for lignin [43]. The reagent, including lithium bromide (LiBr, Xiya reagent company, Shandong, China), hydrochloric acid (HCl, Guoyao reagent company, Shanghai, China), organic solvent N,N-Dimethylformanmide (DMF, Coolaber reagent company, Beijing, China), γ-valerolactone (GVL, Guoyao reagent company, Shanghai, China), tetrahydrofuran (THF, Macklin reagent company, Shanghai, China) and dimethyl sufoxide (DMSO, Solarbio reagent company, Beijing, China) were in chemical reagent grade. The standard substance furfural and 5-HMF were purchased from ANPEL Laboratory Technologies (Shanghai, China) Inc with 99% purity. 

### 3.2. The Conversion of Corn Stover to Furfural and 5-HMF by Acidic LiBr Treatment

The acidic LiBr treatment of corn stover was performed as described before [19,22]. In brief, the crushed corn stover was mixed with the acidic LiBr solution in a 100 mL pressure vessel. A total of 5 wt% catalyst ZSM-5 to the feedstock was adopted. Then the specific co-solvent was added to the reactor. The slurry was sufficiently mixed in the reactor. An oven was used to heat with a rate of 2 °C/min to 140 °C and kept at 140 °C for 150 min. After the reaction, the reactor was carried out from the oven and immersed in tap water to cool down. The supernatant was collected by filtrating with a sand core funnel and stored at 4 °C for further analysis. 

For the recovery of THF and lignin, the reaction solution was added to the suction glass bottle to distill THF in a vacuum by a vacuum pump. The solution was agitated with a magnetic stir, and the THF was boiled at room temperature and collected. After the removal of THF, the extracted lignin was precipitated in the bottom, and it was separated from the solution by centrifugation. Then, the residual lignin was rinsed with water and filtrated through glass fiber filter paper, and dried in a vacuum.

### 3.3. Analyzation of the Products

The concentration of furfural and 5-HMF were quantified by high-performance liquid chromatography (HPLC, UltiMate3000, Thermo Fisher, Waltham, MA, USA) equipped with an ultraviolet detector. The column of Hypersil GOLD TM C18 (Thermo Fisher, Waltham, MA, USA) was used. The column was kept at 30 °C. The mobile phase was methanol to water in a certain gradient. The flow rate was 0.3 mL/min. The yield of furan aldehydes was calculated by the above equations, in which the yields of furan aldehydes are defined as the mass of furfural or 5-HMF formed accounting for the theoretical mass that can be produced from hemicelluloses or cellulose [10]:$$\mathrm{Furfural}\;\mathrm{yield}\;\left(\mathrm{wt}\%\right)=\frac{\mathrm{concentration}\;\mathrm{of}\;\mathrm{furfural}\;\times \mathrm{mass}\;\mathrm{of}\;\mathrm{liquid}\times 1.375}{\mathrm{corn}\;\mathrm{stover}\times \mathrm{fraction}\;\mathrm{of}\;\mathrm{hemicellulose}\;}$$
$$5-\mathrm{HMF}\;\mathrm{yield}\;\left(\mathrm{wt}\%\right)=\frac{\mathrm{concentration}\;\mathrm{of}\;5-\mathrm{HMF}\;\times \mathrm{mass}\;\mathrm{of}\;\mathrm{liquid}\times 1.286}{\mathrm{corn}\;\mathrm{stover}\times \mathrm{fraction}\;\mathrm{of}\;\mathrm{cellulose}\;}$$

### 3.4. Analysis Method

The functional groups of the residual lignin and ZSM-5 were analyzed using Fourier Transform Infrared Spectroscopy (FTIR, Nexus, Thermo Nicolet, Thermo Fisher Scientific, Waltham, MA, USA) with KBr pellets at a range of 400–4000 cm^−1^. The molecular weight (Mw and Mn) of lignin was determined using gel permeation chromatography (GPC, Agilent 1260 HT, U.K.) with dimethylformamide [10,36]. The morphology of ZSM-5 was observed by TEM using JEM 2100 (JEOL, Tokyo, Japan) with an accelerating voltage of 30 kV. The Bet surface for ZSM-5 was measured by using a physisorption analyzer ASAP 2460 (Micromeritics, Atlanta, GA, USA).

## 4. Conclusions

This study reported a green and practical strategy to produce furfural from corn stover by a one-pot reaction with an acidic lithium bromide solution. By investigating various parameters, a decent yield of furfural and 5-HMF was produced in high solid loading (10 wt% solid loading) with industrialized catalyst ZSM-5 zeolite and co-solvent THF at 140 °C for 200 min. During the reaction, the lignin was extracted from the aqueous phase with lower molecular weight and abundant active groups. Moreover, the catalyst and THF were easily recycled with art-known methods. Thus, that method may be a feasible pathway to commercial application.

## Figures and Tables

**Figure 1 ijms-23-14901-f001:**
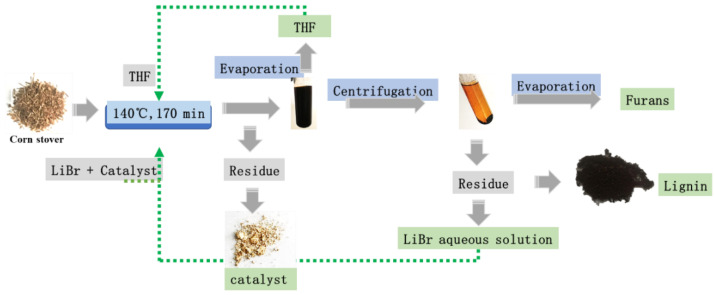
The integrated protocol to apply the method in biorefinery.

**Figure 2 ijms-23-14901-f002:**
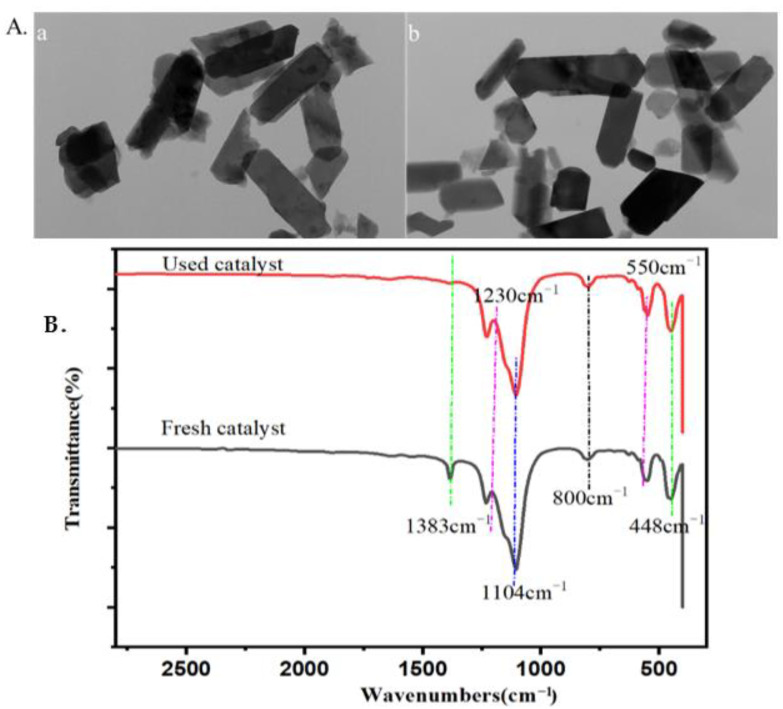
The properties of ZSM-5 before and after the reaction. (**A**) The TEM images of ZSM-5 before (**a**) and after (**b**) reaction, and (**B**) the FTIR spectrum of ZSM-5.

**Figure 3 ijms-23-14901-f003:**
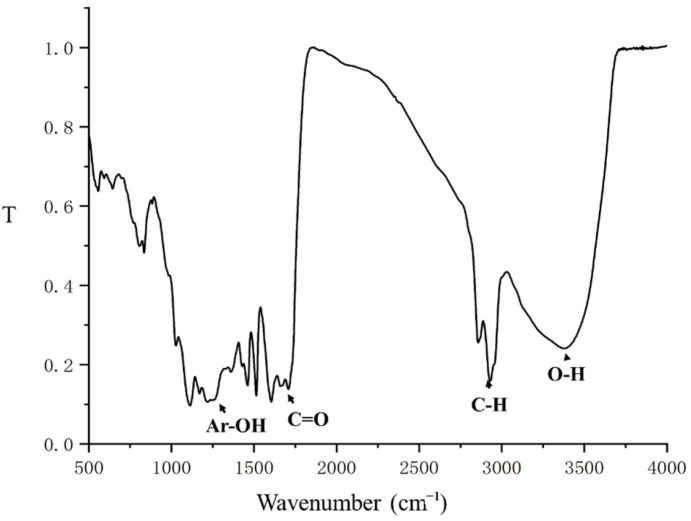
The FTIR spectrum of residual lignin.

**Table 1 ijms-23-14901-t001:** The yield of furfural and 5-HMF from corn stover during acidic LiBr pretreatment. The furfural and 5-HMF in the table were the ratios of the actual yield of furfural and 5-HMF to the theoretical yield.

Substrate Loading (%)	LiBr (%)/Volume	HCl (M)	Catalyst (%)	Organic Phase	Temperature (°C)	Time (min)	Furfural (%)	5-HMF (%)
10	60/10 mL	0.1	-	-	140	120	0.91	3.41
10	60/10 mL	0.5	-	-	140	150	1.72	0.00

**Table 2 ijms-23-14901-t002:** Conversion of corn stover to furfural and 5-HMF by inhibiting the humins formation with the addition of DMF and commercial catalyst ZSM-5.

Substrate Loading (wt%)	LiBr (g/mL)/Volume	HCl (M)	Catalyst (%)	Organic Phase/Volume	Temperature (℃)	Time (min)	Furfural (%)	5-HMF (%)
10	1.2/5 mL	0.1		DMF/5 mL	140	120	-	-
10	1.2/5 mL	0.1	ZSM-5	DMF/5 mL	140	120	4.36	3.05
10	1.2/5 mL	0.5		DMF/5 mL	140	150	31.17	17.15
10	1.2/5 mL	0.5	ZSM-5	DMF/5 mL	140	150	36.00	19.40

**Table 3 ijms-23-14901-t003:** The ratio of the actual yield of furfural and 5-HMF to the theoretical yield with various acid concentrations, the lithium bromide concentration, the reaction temperature, the type of organic, and the ratio of volumetric organic to the aqueous phase.

Test	Substrate Loading (wt %)	LiBr (g/mL)/Volume	HCl (M)	Catalyst (%)	Organic Phase/Volume	Temp (℃)	Time (min)	Furfural (%)	5-HMF (%)
1-1	10	1.2/4 mL	0.5	ZSM-5	DMF/5 mL	140	150	1.47	0.44
2-1	10	1.2/3 mL	0.5	ZSM-5	DMF/5 mL	140	150	1.04	0
3-1	10	1.2/2.5 mL	0.5	ZSM-5	DMF/5 mL	140	150	0.99	0
4-1	10	1.2/1.5 mL	0.5	ZSM-5	DMF/5 mL	140	150	0.98	0
1-2	5	1.2/5 mL	0.5	ZSM-5	DMF/5 mL	140	150	35.49	22
2-2	20	1.2/5 mL	0.5	ZSM-5	DMF/5 mL	140	150	19.65	9.01
1-3	10	1.2/5 mL	0.5	ZSM-5	GVL/ 5 mL	140	150	2.31	0
2-3	10	1.2/5 mL	0.5	ZSM-5	THF/5 mL	140	150	38.1	19.64
3-3	10	1.2/5 mL	0.5	ZSM-5	DMSO/5 mL	140	150	1.29	0.18
1-4	10	1.2/5 mL	0.5	ZSM-5	THF/10 mL	140	150	62.81	29.46
2-4	10	1.2/5 mL	0.5	ZSM-5	THF/15 mL	140	150	74.49	34.94
3-4	10	1.2/5 mL	0.5	ZSM-5	THF/20 mL	140	150	66.68	27.17
1-5	10	1.2/5 mL	0.5	ZSM-5	DMF/5 mL	140	170	39.5	17.35
2-5	10	1.2/5 mL	0.5	ZSM-5	DMF/5 mL	140	200	50.26	15.56
3-5	10	1.2/5 mL	0.5	ZSM-5	DMF/5 mL	140	220	33.6	9.86
6	10	1.2/5 mL	0.5	ZSM-5	THF/15 mL	140	200	100	40.71
7	10	1.2/5 mL	0.5	Reused ZSM-5	THF/15 mL	140	200	100	40.68

**Table 4 ijms-23-14901-t004:** The production of the furfural (Fur) and 5-HMF from corn stover with different reaction systems in which the yield of furfural and HMF was the ratio of the actual yield of product to the theoretical yield.

System	Catalyst	Solid Loading (wt%)	Temp (°C)	Time (min)	5-HMF (%)	Furfural (%)	Ref
LiBr-H_2_O-THF	ZSM-5	10	140	200	40.7	100.0	This paper
GVL-H_2_O	SO_3_H-NG-C	1.3	190	80	17.0	53.0	[34]
H_2_O-MIBK	Ch_15_-AgPW	3.3	170	180	25.6	72.2	[34]
Toluene-NaCl-DMSO	SO_42_−/Sn-TRP	2	190	180	33.2	77.8	[34]
GVL-H_2_O	C-Co-S	1	170	30	-	~70	[34]
GVL-H_2_O	SAPO-18	2.5	205	40	-	95.1	[34]

**Table 5 ijms-23-14901-t005:** The BET surface area, pore size, and acid content of catalyst ZSM-5 before and after LiBr reaction.

Catalyst	BET Surface Area (m^2^/g)	Pore Size (nm)	Total Acid Content (μmol/g)
Before	310	4.11	378
After	308	4.05	370

## Data Availability

The data presented in this study are available on request from the corresponding author.

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
