# Peer review of "Transformation of Corn Stover into Furan Aldehydes by One-Pot Reaction with Acidic Lithium Bromide Solution"

_ijms, 2022, doi:10.3390/ijms232314901_

Round 1
Reviewer 1 Report
The article presents investigation of the catalyst after reaction. The texture characteristics and particle morphology are given. Does the phase composition of the catalyst changes after the reaction? Did the authors determine the carbon content after the reaction?
3.1 line 358 The composition of corn stover is presented. What is included in the remaining 36.33%?
3.1 line 359–361 The names of the manufacturers of the reagents used must be given.
3.1 line 362 The chemical purity of furfural and 5-HMF must be reported.
The paper presents data obtained using the TEM, FTIR, and BET methods, but there is no description of these methods. A description should be provided in section 3.
line 395 400-4000cm-1 -1 is superscript
Reviewer 2 Report
The written English must be improved before the scientific content can be reviewed
Author Response
For reviewer 2:
Point 1: The written English must be improved before the scientific content can be reviewed
Response: The English language of the article has been improved in the revised manuscript.
Please see the attachment.

Reviewer 3 Report
Title: Transformation of Corn Stover into Furan Aldehydes by One-Pot Reaction with Acidic Lithium Bromide Solution
· Comments
1. The submitted article mainly discusses a recyclable and practical strategy to product furfural from corn stover by one-pot reaction with acidic lithium bromide solution without pretreatment and enzymolysis. The article is well written considering explanation of the subject, however there are some flaws which needs to be improved.
2. English correction is needed in many sections particularly abstract needs to be reconstructed. At several places, grammatical corrections are required.
3. Why did author used specifically corn stover only and no other waste substrate.
4. Why authors have selected LiBr ? In fact some other cheaper catalyst options could have been tried. Please explain.
5. It seems reactions were carried out at higher temperatures only. Have authors tried reactions at lower temperatures?
6. Also, author can include recent relevant references related to lignocellulosic biomass and lignin valoraization in the introduction section e.g.
· Singhvi MS, Gokhale DV. Lignocellulosic biomass: hurdles and challenges in its valorization. Applied microbiology and biotechnology. 2019 Dec;103(23):9305-20.
· M Singhvi, BS Kim. Lignin valorization using biological approach.Biotechnology and Applied Biochemistry 68 (3), 459-468.

Author Response
For reviewer 3:
Comments
Point 1:1. The submitted article mainly discusses a recyclable and practical strategy to product furfural from corn stover by one-pot reaction with acidic lithium bromide solution without pretreatment and enzymolysis. The article is well written considering explanation of the subject, however there are some flaws which needs to be improved.
Response: Thank you for your kind and patient comments on the manuscript.
Point 2:2. English correction is needed in many sections particularly abstract needs to be reconstructed. At several places, grammatical corrections are required.
Response: The English language of the article has been revised and the abstract has been reconstructed in the revised manuscript.
Point 3:3. Why did author used specifically corn stover only and no other waste substrate.
Response: corn stover is the main agricultural wastes with the annual output for 247.3 million tons in China. In addition, corn stover is widely available especially in Shandong province. So, corn stover as a typical agricultural wastes is used as the substrate in this study.
Point 4:4. Why authors have selected LiBr ? In fact some other cheaper catalyst options could have been tried. Please explain.
Response: LiBr is selected in this study, because concentrated LiBr hydrate (CLBH, 60 wt%) acts not only as the catalyst, but also as a green solvent/medium to dissolve solid substrate during biorefinery. In the biorefinery of lignocellulose, dissolving of the substrate especially the crystalline cellulose is able to promote the catalytic process. The hydrated lithium cations in CLBH can strongly coordinate with hydroxyl groups in cellulose, thus the extensive intra- and inter- chain hydrogen-bond network was disrupted, and the compact crystalline cellulose can be swollen and even dissolved in CLBH.
Point 5:5. It seems reactions were carried out at higher temperatures only. Have authors tried reactions at lower temperatures?
Response: thank you for your comments, but higher treatments intensity was necessary in this reaction, because two steps are needed for HMF or furfural production from lignocellulose, including hydrolysis of polysaccharide to monosaccharide and further dehydration of monosaccharides to HMF and furfural. The results of this study indicated that only 0.91 % of furfural was yielded and 50.89 % xylose was residual in the supernatant at 140℃ for 120 min. To promote the dehydration process of xylose, higher treatment intensity was needed such as raising temperatures, prolonging the reaction time, or adding HCl. So, the temperatures lower than 140℃ did not tried in this study.
Point 6:6. Also, author can include recent relevant references related to lignocellulosic biomass and lignin valoraization in the introduction section e.g.
- Singhvi MS, Gokhale DV. Lignocellulosic biomass: hurdles and challenges in its valorization. Applied microbiology and biotechnology. 2019 Dec;103(23):9305-20.
- M Singhvi, BS Kim. Lignin valorization using biological approach.Biotechnology and Applied Biochemistry 68 (3), 459-468.
Response: thank you for your advice, and the relevant references were added in the introduction section.
Please see the attachment.

Reviewer 4 Report
The article named Transformation of corn stover into furan aldehydes by one-pot reaction with acidic lithium bromide solution is focused on the determination of the suitable conditions of the decomposition of lignocellulosic biomass to furfural and 5-hydroxymethyl furfural with aim to reached the highest yields of the products. Between main studied conditions were concentration of LiBr, substrate loading, type and amount of organic solvents and time of decomposition. The work was focused also on the characterization of solids catalysts and reusability catalyst. The work is very interesting with interesting results Moreover, it presented, that a lot of work was done. Unfortunately, the article is very hard to read, therefore I suggested better division into chapters where the partly conclusion will be in every chapter. It will help to reader better orientation on article. The introduction is very long. Moreover, the focusing of own work on the end of the introduction missing. Then, the yield of LiBr missing. Was concentration of LiBr analyzed before and after reaction? Was Li salts formed of organic compounds? Why was used of HCl, when ZSM-5 was used as heterogenous solids acid? What role has ZSM-5 in process? On line 215 should be reaction time instead of temperature. On base of the total evaluation i concluded, that the article has not to be published in this formed. I suggest major revision.
Author Response
For reviewer 4:
Point 1: The article named Transformation of corn stover into furan aldehydes by one-pot reaction with acidic lithium bromide solution is focused on the determination of the suitable conditions of the decomposition of lignocellulosic biomass to furfural and 5-hydroxymethyl furfural with aim to reached the highest yields of the products. Between main studied conditions were concentration of LiBr, substrate loading, type and amount of organic solvents and time of decomposition. The work was focused also on the characterization of solids catalysts and reusability catalyst. The work is very interesting with interesting results Moreover, it presented, that a lot of work was done. Unfortunately, the article is very hard to read, therefore I suggested better division into chapters where the partly conclusion will be in every chapter. It will help to reader better orientation on article. The introduction is very long. Moreover, the focusing of own work on the end of the introduction missing.
Response: thank you for your comments, and the introduction section was modified in the revised manuscript.
Point 2: Then, the yield of LiBr missing. Was concentration of LiBr analyzed before and after reaction? Was Li salts formed of organic compounds?
Response: thank you for your comments, and the concentration of LiBr was not analyzed after this reaction, but we detected the catalytic ability of the reused LiBr in our previous study. The results indicated that the catalytic ability was slightly influenced in the first three recycling process. So the concentration of LiBr is supposed changed a little in the first three recycling, but in the following reaction cycles, fresh LiBr was needed to supplement in the reaction.
The LiBr reagent was purchased from Xiya reagent company, and the forming method to Li salt was not investigate.
Point 3: Why was used of HCl, when ZSM-5 was used as heterogenous solids acid? What role has ZSM-5 in process?
Response: in the reaction, HCl and ZSM were both as the acidic catalysts to hydrolyze lignocellulose to monosaccharides followed by the generation of furan aldehydes. Compared with the reaction with HCl, addition of commercial catalyst ZSM-5 can further enhance the yield of furfural.
Point 4: On line 215 should be reaction time instead of temperature.
Response: thank you for your comments, and “reaction time” was corrected on line 215.
On base of the total evaluation i concluded, that the article has not to be published in this formed. I suggest major revision.
Please see the attachment

Reviewer 5 Report
The paper entitled by “ Transformation of Corn Stover into Furan Aldehydes by One-Pot reaction with Acidic Lithium Bromide Solution” with the reference Ijms-1982069 for International Journal of Molecular Sciences is focused in the production of furanic aldehydes from raw biomass which have important applications in several industry markets. The theme is of great importance concerning the emergent need of sustainable chemicals, however I don’t think it is suitable for publication in a Q1 magazine with a high impact factor because it is has a poor english writing language even in the abstract, lack of explanations for certain criterions, lots of repetitive observations and a confusing structure of results presentation.
Here are my reasons:
1) Examples of the bad english:
1.1- In the abstract the authors wrote “ Here, this context reported a recyclable and practical strategy to product furfural…”;
1.2- “ By inhibiting the dehydration reaction, the high yield of furan aldehydes were obtained, 2.94 g/L of furfural and 2.78 g/L of 5-hydroxymethyl furfural (5-HMF) yielded with high solid loading (10 wt% corn stover loading).
1.3- In the first sentence of the introduction the authors wrote et al as a kind of etc? For me Et al. is used when citing a certain reference… Not as an etc…
1.4- In the introduction “ ….mainly composed of xylan, is the readily available sugar in lignocellulose…”.
1.5- In the indtroduction yet the authors wrote “ Furfural and furfural derivatives, as an industrial chemical platform to produce renewable chemicals and biofuels, their demand would increase”, seems like it misses some connection and is incomplete.
1.6- On line 73 the authors start a sentece with an And?
2) In the abstract the authors say that almost 100 % of hemicellulose was transformed to furfural and 40.71 % of cellulose was transformed to 5-HMF.
But 8.35 g of HMF was obtained from 32.56 g of cellulose, which corresponds to ca. 26% of cellulose and not 40.71 %.
3) The authors wrote that the industrial furfural production process is based on H2SO4 catalysed reaction. However they don’t say where it is produced and don’t give a source to whether they took this information neither.
4) The authors wrote “ For example, by using pulp pre-hydrolysate derived from prehydrolysis of wood, 10 hours H2SO4 treatment was needed to obtain relatively high yield of furfural (78.6%) [15]. The low furfural yield….”.
This sentence is confusing. In the first part of this sentence you say the furfural was obtained in 79 % and just afterwards its writen that the furfural yield is low preventing the commercial transformation of raw biomass to furan aldehydes. But 79 % is a high yield. So its confusing.
5) The authors wrote “ … in which the degradation products of lignocellulose, such as glucose, furfural and 5-HMF…”.
This is also confusing. it seems like the aim of this work was to obtain furanic aldehydes: Furfural and 5-hydroxymethylfurfural. However here in this sentence it seems like they are considered as degradation products..
6) The authors wrote “ Thus, as a low toxicity and recyclable strategy in the utilization of raw crop residues, acidic LiBr treatment…”.
I dont understant the sequence of this sentence here. After saying that recent studies showed that the acidic LiBr reaction lead to the formation of high amounts of humnis and low sugars, then the authors say it was used to limit the formation of humins? I think that the introduction must suffer a big improvement.
7) The results and discussion part is very poorly organized and discussed. The authors discuss yields enhancements which are in my opinion barely the same and within the experimental error. For example in line 96 they give importance to an improvement of 1.72 % yield! The same in line 123 where the authors considered an increase of 5 % also significant. Or in line 141 where the authors even wrote “ Sharply reduced” to a change from 1.47 to 0.98 %.
8) The authors say that there is a large amount of humins but they do not quantify them.
9) In the legend of Table 1 the authors wrote “ The furfural and 5-HMF in table were the yield of furfural and 5- HMF to the theoretical yield”.
I don´t understand this. The given yields are the yields in relation to the theoretical?
10) In Table 2, the percentage of the substrate loading should be specified if it is molar or mass.
Still in Table 2, the 120 value (120/5 mL) is %? It is possible to be more than 100 %?
11) In line 125 the authors put the reference 11. Why? Wasn’t this result obtained in this work?
12) In line 133 the authors changed the substrate loading from 5 % to 20 % but they do not mention if it is molar or mass.
13) In line 152 the authors wrote “ 1.2 g/mL LiBr aqueous phase was elected in this reaction”. This is repeated in lines 114-115.
14) In line 154, the authors wrote “ melt temperature”.
What the authors mean by at the melt temperature? Is it melting temperature of LiBr?
15) In line 155 the authors wrote “ solid loading”. But what solid? The substrate or catalyst?
16) In line 156 the authors wrote “ high yield of product”. But what product»? Furanic aldehydes? Furfural or 5-HMF?
17) The discussion between lines 161-170 is very confusing. It should be re-written.
18) In Table 3, on the catalyst column the authors wrote re-freshed ZSM-5 for the test 7, but shouldn’t it be reused ZSM-5?
19) Also in Table 3, on test 6 and 7 , the total yield (when adding the Furfural and 5-HMF yields) are higher than 100 % which is impossible.
20) In line 176 the authors wrote “ With the optimal LiBr concentration and high loading of substrate LiBr, cellulose and hemicellulose were able to be saccharified…”. But I don’t understand this sentence.. Maybe it should be re-written.
21) The sentence between 187-189 is confusing because the ca. 36 % of furfural yield in the case of DMF is obtained by using 5 wt% of substrate while the 38 % obtained in the presence of THF is obtained using 10 wt%. So they are not compared in the same conditions. Fair comparisons should be done under the same conditions. Otherwise you cannot compare them.
22) In line 191, the authors wrote “ especially DMSO”. Why especially? One gives 1.29 and the other 2.39. Do the authors think it makes a significant difference?
23) The sentence in lines 197-198 is confusing. It should be clarified.
24) The text from lines 202-209 is very repetitive with lines 178-180.
25) Table 3 is very confusing and very annoying to read. it should be divided by the type of parameter to be discussed to turn it less painful to read
26) In line 219 the authors wrote “ with 10 wt% corn stover loading”.
All the document, it is very repetitive.. Why the authors say again with 10 wt% corn stover in line 219 if on lone 217 you already mentioned it was 10 wt%?
27) In line 228 the authors really mean conversation? Or is it conversion?
28) In line 231-232 (sulfonated carbon microspheres C-Co-S) as catalyst is also repetitive as in line 230 they already wrote the catalyst used.
29) In line 233 the 2% solid are related to what? the substrate? Or the catalyst? Many things are solids but have different roles.
30) Concerning the discussion on Table 4, the authors only report the results obtained in a certain reference. Do not have any valuable comment on each of their results.
31) In line 246, the authors wrote “ High solid loading (10 wt%)”. What solid? Substrate? Catalyst?
32) In line 248, the authors wrote that the results reported are obviously superior to the reported ones but that depends on the effective results you actually obtained because certainly 140.7 % total product of HMF and Fur is not possible (as included in Table 4).
33) The authors measured the acid content, but they do not refer how they were measured (if it was by FT-IR pyridine, TPD of NH3…). That should be specified.
34) The authors also do not mentioned if the used catalyst characterised just after separated from the reaction mixture or after calcination?
35) The cornstover used contains Mn? (as referred in line 293). Maybe the composition of the cornstover should be included.
36) An illustrative figure would help to visualize the mechanism explained in lines 306-313.
37) The authors said they studied the reusability of the catalyst, but they only did a second run. At least 3 to 4 recycling runs should be reported.
38) The authors wrote in line 331 “ furan aldehydes was evaporated”. Does This means that the products of interest in this system are not recovered? Therefore what is the interest of this work?
39) In line 336 the authors wrote “ suppresses the formation of humins”. But the authors say several times that humins were formed even in the presence of organic solvents.
40) In lines 341, 343 and 346 the authors wrote that there were obtained 7.86 g furfural, 8.35 g of HMF, 12.29 g xylose, 29.30 g de glucose. I dont understand how the authos arrived to these values. Can the authors explain?
Concerning the yield of HMF, how the authors calculate the yield to be of 40.71 %? if it is obtained 8.35 g of HMF which is the maximum possible to be obtained from 100 g of corn stover why it is not considered to be 100% of the theoretical value as well?
41) In lines 390, Why the formula to obtain the Furfural yield is multiplied by 1.375?
42) What was the brand and equipment for the GPS results?
Author Response
For reviewer 5:
The paper entitled by “Transformation of Corn Stover into Furan Aldehydes by One-Pot reaction with Acidic Lithium Bromide Solution” with the reference Ijms-1982069 for International Journal of Molecular Sciences is focused in the production of furanic aldehydes from raw biomass which have important applications in several industry markets. The theme is of great importance concerning the emergent need of sustainable chemicals, however I don’t think it is suitable for publication in a Q1 magazine with a high impact factor because it is has a poor english writing language even in the abstract, lack of explanations for certain criterions, lots of repetitive observations and a confusing structure of results presentation.
Here are my reasons:
Point 1: 1) Examples of the bad english:
- In the abstract the authors wrote “Here, this context reported a recyclable and practical strategy to product furfural…”;
Response: the abstract section was reconstructed according to reviewers’ comments. And the sentence in the abstract was replaced by “In this study, a recyclable and practical method was explored for preparation of furfural… “ in the revised paper.
Point 2:“ By inhibiting the dehydration reaction, the high yield of furan aldehydes were obtained, 2.94 g/L of furfural and 2.78 g/L of 5-hydroxymethyl furfural (5-HMF) yielded with high solid loading (10 wt% corn stover loading).
Response: the sentence in the abstract was replaced by “So, dehydration reaction were inhibited in this study and the high-yield of furan aldehydes were obtained, in which 2.94 g/L of furfural and 2.78 g/L of 5-hydroxymethyl furfural (5-HMF) were generated with high solid loading (10 wt% corn stover loading), the present of commercial catalyst ZSM-5 and co-solvent tetrahydrofuran (THF) at 140 ℃ for 200 min. “ in the revised paper.
Point 3:In the first sentence of the introduction the authors wrote et al as a kind of etc? For me Et al. is used when citing a certain reference… Not as an etc…
Response: thank you for your advice, and the “et al.” was corrected to “etc.” in the revised manuscript.
Point 4:In the introduction “ ….mainly composed of xylan, is the readily available sugar in lignocellulose…”.
Response: the sentence in the introduction was replaced by “in which the major component of hemicellulose is xylan and that is easily hydrolyzed during moderate severity acidic treatment “ in the revised paper.
Point 5:In the indtroduction yet the authors wrote “ Furfural and furfural derivatives, as an industrial chemical platform to produce renewable chemicals and biofuels, their demand would increase”, seems like it misses some connection and is incomplete.
Response: the sentence in the introduction was replaced by “The demand of furfural and furfural derivatives would increase, since they act as an industrial chemical platform to produce renewable chemicals and biofuels “ in the revised paper.
Point 6:On line 73 the authors start a sentece with an And?
Response: the “And” was deleted in the revised manuscript.
Point 7: In the abstract the authors say that almost 100 % of hemicellulose was transformed to furfural and 40.71 % of cellulose was transformed to 5-HMF.
But 8.35 g of HMF was obtained from 32.56 g of cellulose, which corresponds to ca. 26% of cellulose and not 40.71 %.
Response: in this reaction, 100 g corn stover is composed of 13.97 g hemicellulose, 32.56 g cellulose and 17.14 g lignin. In the first step, the cellulose was transformed to glucose (29.30 g). In the second step of this reaction, the theoretical yield for HMF is 20.51 g. so, the actual yield of 5-HMF is 40.71 % (8.35 g vs 20.51 g) based on the theoretical yield.
Point 8: The authors wrote that the industrial furfural production process is based on H2SO4 catalysed reaction. However they don’t say where it is produced and don’t give a source to whether they took this information neither.
Response: this sentence was written based on the reference 6, as showed in the manuscript.
Point 9:The authors wrote “ For example, by using pulp pre-hydrolysate derived from prehydrolysis of wood, 10 hours H2SO4 treatment was needed to obtain relatively high yield of furfural (78.6%) [15]. The low furfural yield….”.
This sentence is confusing. In the first part of this sentence you say the furfural was obtained in 79 % and just afterwards its writen that the furfural yield is low preventing the commercial transformation of raw biomass to furan aldehydes. But 79 % is a high yield. So its confusing.
Response: it was sorry for the confusing expression and the ~79 % yield is high for lignocellulose substrate, however, energy consumption was high in the process. To avoid the confusing, this sentence has been adjusted to “The highly energy intensive and relatively low furfural yield (vs model sugar) ” in the revised manuscript.
Point 10:The authors wrote “ … in which the degradation products of lignocellulose, such as glucose, furfural and 5-HMF…”.
This is also confusing. it seems like the aim of this work was to obtain furanic aldehydes: Furfural and 5-hydroxymethylfurfural. However here in this sentence it seems like they are considered as degradation products..
Response: the sentence has been modified to “which were easily crosslinked with glucose and xylose, or self-polymerized to form undesired products, such as humins” in the revised paper.
Point 11:The authors wrote “ Thus, as a low toxicity and recyclable strategy in the utilization of raw crop residues, acidic LiBr treatment…”.
I dont understant the sequence of this sentence here. After saying that recent studies showed that the acidic LiBr reaction lead to the formation of high amounts of humnis and low sugars, then the authors say it was used to limit the formation of humins? I think that the introduction must suffer a big improvement.
Response: thank you for your advice, and the introduction has been improved according to reviewers’ comments. Recent studies showed that the acidic LiBr reaction leads to the formation of high amounts of humnis and low sugars. So, this study was carried out to explore the methods to reduce the formation of humins and to increase the furfural yield.
Point 12:The results and discussion part is very poorly organized and discussed. The authors discuss yields enhancements which are in my opinion barely the same and within the experimental error. For example in line 96 they give importance to an improvement of 1.72 % yield! The same in line 123 where the authors considered an increase of 5 % also significant. Or in line 141 where the authors even wrote “ Sharply reduced” to a change from 1.47 to 0.98 %.
Response: thank you for your advice, and the results and discussion part has been improved according to reviewers’ comments. And the “Sharply” was deleted in the revised paper to avoid confusing.
Point 13:The authors say that there is a large amount of humins but they do not quantify them.
Response: in this reaction, the humin was formed mixed with the residual solid to make the residues in black color, so it is hard to quantify the amount of humins.
Point 14:In the legend of Table 1 the authors wrote “ The furfural and 5-HMF in table were the yield of furfural and 5- HMF to the theoretical yield”.
I don´t understand this. The given yields are the yields in relation to the theoretical?
Response: the furfural and 5-HMF in table were the ratio of the actual yield of furfural and 5- HMF to the theoretical yield. The sentence was modified to “The furfural and 5-HMF in table were the ratio of the actual yield of furfural and 5- HMF to the theoretical yield” for clarity.
Point 15:In Table 2, the percentage of the substrate loading should be specified if it is molar or mass.
Still in Table 2, the 120 value (120/5 mL) is %? It is possible to be more than 100 %?
Response: thank you for your comments, and the “wt%” was added in the substrate loading. In addition, the 120 % LiBr means that “1.2 g/mL LiBr aqueous phase was added to keep the certain concentration of LiBr during reaction, because of the high solubility of LiBr in organic and aqueous phases.”
Point 16: In line 125 the authors put the reference 11. Why? Wasn’t this result obtained in this work?
Response: the result was obtained in this work, and the reference 11 was put in this sentence to clarify the catalytic ability of ZSM-5. In the revised manuscript, the reference 11 was deleted.
Point 17: In line 133 the authors changed the substrate loading from 5 % to 20 % but they do not mention if it is molar or mass.
Response: thank you for your comment, and the “%” was replaced by “wt%” in the revised paper.
Point 18: In line 152 the authors wrote “ 1.2 g/mL LiBr aqueous phase was elected in this reaction”. This is repeated in lines 114-115.
Response: the “1.2 g/mL LiBr aqueous phase was elected in this reaction” in line 114-115 was deleted in the revised paper.
Point 19: In line 154, the authors wrote “ melt temperature”.
What the authors mean by at the melt temperature? Is it melting temperature of LiBr?
Response: the melt temperature is the dissolved temperature for LiBr in deionized water.
Point 20: In line 155 the authors wrote “ solid loading”. But what solid? The substrate or catalyst?
Response: the solid loading means the solid loading of substrate and that was adjusted in the revised manuscript.
Point 21:In line 156 the authors wrote “ high yield of product”. But what product»? Furanic aldehydes? Furfural or 5-HMF?
Response: the high yield of product means “Furfural and 5-HMF” and that was adjusted in the revised manuscript.
Point 22:The discussion between lines 161-170 is very confusing. It should be re-written.
Response: thank you for your advice, and that section were re-written in the revised paper.
Point 23:In Table 3, on the catalyst column the authors wrote re-freshed ZSM-5 for the test 7, but shouldn’t it be reused ZSM-5?
Response: the “re-freshed” were corrected to “reused” in the revised paper.
Point 24:Also in Table 3, on test 6 and 7 , the total yield (when adding the Furfural and 5-HMF yields) are higher than 100 % which is impossible.
Response: the ratios of furfural and 5-HMF yield were calculated respectively, in which the yield of furfural in table was the ratio of the actual yield of furfural to the theoretical yield of furfural and that for 5-HMF was calculated according to the theoretical yield of 5-HMF.
Point 25:In line 176 the authors wrote “ With the optimal LiBr concentration and high loading of substrate LiBr, cellulose and hemicellulose were able to be saccharified…”. But I don’t understand this sentence.. Maybe it should be re-written.
Response: The optimal reaction condition was carried out, and cellulose and hemicellulose were able to be saccharified to produce glucose and xylose, then the monosaccharides were further dehydrated to form furan aldehydes. That was re-written in the revised paper.
Point 26:The sentence between 187-189 is confusing because the ca. 36 % of furfural yield in the case of DMF is obtained by using 5 wt% of substrate while the 38 % obtained in the presence of THF is obtained using 10 wt%. So they are not compared in the same conditions. Fair comparisons should be done under the same conditions. Otherwise you cannot compare them.
Response: thank you for your advice, and the comparisons were deleted in the revised paper.
Point 27:In line 191, the authors wrote “ especially DMSO”. Why especially? One gives 1.29 and the other 2.39. Do the authors think it makes a significant difference?
Response: thank you for your advice, and the “especially DMSO” was deleted in the revised paper.
Point 28:The sentence in lines 197-198 is confusing. It should be clarified.
Response: the sentence was rewritten to “The ratios of organic solvent-aqueous phase were set as 1:1, 2:1, 3:1 and 4:1 (THF to LiBr aqueous phase, v/v) in the biphasic reaction” in the revised paper.
Point 29:The text from lines 202-209 is very repetitive with lines 178-180.
Response: the text from lines 202-209 was deleted in the revised paper.
Point 30:Table 3 is very confusing and very annoying to read. it should be divided by the type of parameter to be discussed to turn it less painful to read
Response: thank you for your advice, and the Test name was defined according to the type of parameter, such as the 1-1 to 1-4 were named according to the reaction conditions with different LiBr concentrations, and 2-1 to 2-2 were set according to the parameter of solid loading.
Point 31:In line 219 the authors wrote “ with 10 wt% corn stover loading”.
All the document, it is very repetitive.. Why the authors say again with 10 wt% corn stover in line 219 if on lone 217 you already mentioned it was 10 wt%?
Response: the 10 wt% corn stover in line 219 was deleted in the revised paper.
Point 32:In line 228 the authors really mean conversation? Or is it conversion?
Response: thank you for your advice, the “conversation” were corrected to “conversion” in the revised paper.
Point 33:In line 231-232 (sulfonated carbon microspheres C-Co-S) as catalyst is also repetitive as in line 230 they already wrote the catalyst used.
Response: the “sulfonated carbon microspheres C-Co-S” in line 231-232 was deleted.
Point 34:In line 233 the 2% solid are related to what? the substrate? Or the catalyst? Many things are solids but have different roles.
Response: that means 2 % solid loading for substrate, and “for substrate” were added in the revised paper.
Point 35:Concerning the discussion on Table 4, the authors only report the results obtained in a certain reference. Do not have any valuable comment on each of their results.
Response: the discussion of Table 4 were showed in the lines 223-256, and “ as showed in Table 4” was added in line 224.
Point 36:In line 246, the authors wrote “ High solid loading (10 wt%)”. What solid? Substrate? Catalyst?
Response: “high solid loading (10 wt%) for substrate” was added in the revised paper.
Point 37:In line 248, the authors wrote that the results reported are obviously superior to the reported ones but that depends on the effective results you actually obtained because certainly 140.7 % total product of HMF and Fur is not possible (as included in Table 4).
Response: The ratios of furfural and 5-HMF yield were calculated respectively, in which the yield of furfural in table was the ratio of the actual yield of furfural to the theoretical yield of furfural and that for 5-HMF was calculated according to the theoretical yield of 5-HMF.
Point 38:The authors measured the acid content, but they do not refer how they were measured (if it was by FT-IR pyridine, TPD of NH3…). That should be specified.
Response: TPD of NH3 was used to measure the acid content, that was added in the revised paper.
Point 39:The authors also do not mentioned if the used catalyst characterised just after separated from the reaction mixture or after calcination?
Response: the catalyst was characterized just after separated from the reaction mixture, and that was supplemented in the revised paper.
Point 40:The corn stover used contains Mn? (as referred in line 293). Maybe the composition of the cornstover should be included.
Response: the Mn means the number average molar mass for lignin in line 293.
Point 41:An illustrative figure would help to visualize the mechanism explained in lines 306-313.
Response: thank you for your comment, but this mechanism was already reported in other papers [1], so the illustrative figure was not provided in this paper.
[1] Xing, R.; Qi, W.; Huber, G.W. Production of furfural and carboxylic acids from waste aqueous hemicellulose solutions from the pulp and paper and cellulosic ethanol industries. Energy & Environmental Science 2011, 4.
Point 42:The authors said they studied the reusability of the catalyst, but they only did a second run. At least 3 to 4 recycling runs should be reported.
Response: thank you for your comment, and the catalyst was only reused in a second run in this study to verify the reusability of ZSM-5. So, more recycling runs were not performed, and our further research would do more recycling runs.
Point 43:The authors wrote in line 331 “ furan aldehydes was evaporated”. Does This means that the products of interest in this system are not recovered? Therefore what is the interest of this work?
Response: the furan aldehydes were separated from the reaction system by evaporation. And that sentence was modified to “ the furan aldehydes was recovered in the following by evaporation with boiling point of 97.9 ℃” in the revised paper.
Point 44:In line 336 the authors wrote “ suppresses the formation of humins”. But the authors say several times that humins were formed even in the presence of organic solvents.
Response: “suppresses” was corrected to “reduce” in the revised paper to avoid confusing.
Point 45:In lines 341, 343 and 346 the authors wrote that there were obtained 7.86 g furfural, 8.35 g of HMF, 12.29 g xylose, 29.30 g de glucose. I dont understand how the authos arrived to these values. Can the authors explain?Concerning the yield of HMF, how the authors calculate the yield to be of 40.71 %? if it is obtained 8.35 g of HMF which is the maximum possible to be obtained from 100 g of corn stover why it is not considered to be 100% of the theoretical value as well?
Response: in this reaction, 100 g corn stover is composed of 13.97 g hemicellulose, 32.56 g cellulose and 17.14 g lignin. In the first step, the cellulose was transformed to glucose (29.30 g). In the second step of this reaction, the theoretical yield for HMF is 20.51 g. so, the actual yield of 5-HMF is 40.71 % (8.35 g vs 20.51 g) based on the theoretical yield.
Point 46: In lines 390, Why the formula to obtain the Furfural yield is multiplied by 1.375?
Response: because the transformation factor for hemicellulose to furfural is 0.727, so 1.375 is showed as the inverse of 0.727 in this formula.
Point 47: What was the brand and equipment for the GPS results?
Response: the GPC (Agilent 1260 HT, England) was used in this work and that was added in the revised paper.
Please see the attachment.

Round 2
Reviewer 3 Report
Authors have addressed all issues carefully and revised MS seems to be suitable for publication now.
Best Rgds
Mamata
Author Response
Point 1: Authors have addressed all issues carefully and revised MS seems to be suitable for publication now.
Best Regards,
Mamata
Response: thank you for your comments.
Reviewer 4 Report
Thank you for explanation, which helped me understanding of catalyst stability. Unfortunately, the explanation of the ZSM-5 role was not sufficient. The improve of the furfural and 5 HMF yield is observed with using ZSM-5 as catalysts. The ZSM-5 is solid acid, which it replace strong inorganic acids as catalysts in organic reactions. Therefore, the used of both acid catalysts (HCl and ZSM-5) has to have some point. The explanation could bring the decomposition without HCl, only ZSM-5 and further HCl with different zeolite, which has absolutely different pore size (zeolite BEA). The information about Si/Al molar ration missing. Molar ration Si/Al has influence of the power and concentration of acids sites. The role of ZSM-5 could be in selectivity to furfural and 5HMF, where ZSM-5 will play role of the molecular sieve and catalyst. However, it have to be substantiated of results. The role of ZSM-5 should be explained in article.
Reviewer 5 Report
The revised version for “ Transformation of Corn Stover into Furan Aldehydes by One-Pot reaction with Acidic Lithium Bromide Solution” with the reference Ijms-1982069 for International Journal of Molecular Sciences has been improved and the authors answered to mostly all of the questions, however, I am afraid that this is not yet suitable for International Journal of Molecular Sciences.
(1) The manuscript should yet suffer some improvements on the english language even in the sentences they re-wrote. For example “ the presence of commercial catalyst” instead of present as they wrote? And if they write “ High solid loadig” shouldn’t it be enough to put in parenthesis (10 wt%), instead of writing again 10wt% loading? Also in the re-written sentence “in which the major component of hemicellulose is xylan and that is easily hydrolyzed during moderate severity acidic treatment “ in the revised paper, is not corrected written despite being understandable. The same accounts for other sentences namely the discussed in my previous points 9 and 22. I understood if after the authors explanation but it still is confusing for a general reader to understand it.
(2) Concerning my previous point 7, Thanks for your explanation. But I still think it should be better explained in the text that the 8.35 g of HMF is in relation to the the theoretical yield to be obtained from those 29.30 g of glucose?
(3) Concerning my previous point 8, maybe this ref 6 should be written after this sentence again.
(4) Concerning my previous point 11, I still cannot understand why you used acid LiBr with the aim of limiting the formation of humins if recent studies mention they led to high amounts of humins. Unless you state a big advantage of using LiBr with compensate the formation of humins, it is difficult to accept their use knowing this limitation.
(5) Concerning my previous point 12, In fact you still consider that 0.91 % yield to 1.72 % in an enhancement (it is written in line 104. Frankly its a change in less than 1 %. Do you really consider it an improvement after changing the concentration of HCl significantly from 0.1 to 0.5 M?
(6) Thanks for your reply concerning my point 13 but i think it should be nice to explain this in the main text.
(7) On the answer to my previous point 14, you said that “ the furfural and 5-HMF in table were the ratio of the actual yield of furfural and 5- HMF to the theoretical yield” and that “The sentence was modified to “The furfural and 5-HMF in table were the ratio of the actual yield of furfural and 5- HMF to the theoretical yield” for clarity. However, I dont see this change on the revised manuscript for Table 1. Only in Table 3. The values given in Table 1 are also the ratio in relation to the theoretical yields?
(8) Concerning my previous point 15, Thanks for your answer, but wouldn't it be preferable to put 1.2 g/mL of LiBr in aqueous phase than writing 120 %/5 mL?
(9) Concerning my previous point 24, does it means that all the results presented are in relation to the theoretical including those minor yields of less than 1 or 2 wt%? Why do you present the results in relation to the theoretical yield? It is more difficult to see what yield you are getting in such a way.
(10) Concerning my previous point 35, Do the results of these refs are also in relation to the theoretical value?
(11) Concerning my previous point 37, you answered that “The ratios of furfural and 5-HMF yield were calculated respectively, in which the yield of furfural in table was the ratio of the actual yield of furfural to the theoretical yield of furfural and that for 5-HMF was calculated according to the theoretical yield of 5-HMF”. If thats so, you should put it as legend of Table 4 as well.
(12) Concerning my previous point 41, maybe you should make reference to this reference 1 in the explanation for the mechanism proposal?
Round 3
Reviewer 5 Report
The authors of the revised version for “ Transformation of Corn Stover into Furan Aldehydes by One-Pot reaction with Acidic Lithium Bromide Solution” with the reference Ijms-1982069 for International Journal of Molecular Sciences have clarified me about almost everything unless of one very important detail concerning my previous points 4 and 5.
I understand that LiBr is effective to hydrolyse polysaccharides and that increasing residual time and temperature, the yields of furanic aldehydes increase. But the only thing you had verified was that the humins increased, which does not justify the sentence you wrote that LiBr inhibits the formation of humins. It misses some explanation like " although LiBr has the disadvantage of leading to a significant amount of humins (which you also wrote), it has the advantage of XXX, which compensates this drawback”. Or something like “ Although the presence of humins formed in the presence of LiBr is still high, it is considerable decreased in relation to other common alternatives as XXX or XX”.
Moreoever, I understand that the xylose content was significant reduced, but this reduction was not reflected in the increment of the furanic aldehydes if they only increased in 1 %. This leads to think that the decrease of xylose content was only reflected in an increament on the formation of humins. Which leads to the question in how the LiBr inhinits the formation of humins?
To have my positive answer in this manuscript I would need this answer.
